# Residual-aware health prediction of power transformers via spatiotemporal graph neural networks

**Peng Zeng, Gong Chu** *

School of Intelligent Manufacturing, Yibin Vocational and Technical College, Yibin, Sichuan Province, China

* chugedu@163.com

**Data availability statement:** Corresponding data can be found at the DOI number is as follows https://doi.org/10.5281/zenodo.17075969.

## Abstract

Accurate health state prediction and timely fault detection of power transformers are critical for ensuring the reliability and resilience of modern power systems. This paper proposes a residual-aware spatiotemporal graph neural network (STGNN) framework that jointly models dynamic topological dependencies among multivariate Supervisory Control and Data Acquisition (SCADA) signals and their temporal evolution. The proposed approach constructs time-varying sensor graphs via attention mechanisms to capture evolving inter-variable relationships and applies Chebyshev spectral graph convolution for localized spatial feature aggregation. Temporal dependencies are modeled using gated recurrent units (GRUs) enhanced with residual connections, enabling robust forecasting under nonstationary operating conditions. A health indicator (HI) is derived from node-level prediction residuals, and anomalies are detected using a quantile-based thresholding strategy. Extensive experiments are conducted on a synthetic SCADA dataset simulating five major transformer subsystems—winding, core, cooling, insulation, and tap changer—under both nominal and faulty conditions. Results demonstrate that the proposed STGNN achieves superior forecasting accuracy and significantly outperforms baseline methods in anomaly detection, particularly under noisy and dynamic scenarios. The framework offers a scalable, interpretable, and deployment-ready solution for intelligent condition monitoring in substation automation systems.

## 1 Introduction

Power transformers are critical components in ensuring the reliability and operational stability of modern power grids. Unexpected failures in core subsystems—such as windings, insulation layers, or cooling units—can lead to widespread blackouts and significant economic losses. To mitigate these risks, Supervisory Control and Data Acquisition (SCADA) systems have been widely deployed in substations to enable continuous monitoring of multivariate sensor streams, including winding temperature, core flux density, load current, partial discharge amplitude, oil level, and ambient environmental conditions [1,2].

**Funding:** The author(s) received no specific funding for this work.

**Competing interests:** The authors have declared that no competing interests exist.

These time-series signals contain latent degradation patterns, yet extracting meaningful spatiotemporal dependencies from high-dimensional data remains a significant challenge.

Specifically, this study addresses the practical scenario where transformer health must be *predicted* and *evaluated* under nonstationary operating conditions by jointly modeling (i) spatial couplings among subsystem-related sensor variables and (ii) temporal dynamics in multivariate SCADA sequences, while also providing residual-based indicators for early and interpretable fault detection.

Traditional approaches to transformer condition monitoring primarily rely on statistical modeling and signal processing techniques. Classical methods such as principal component analysis (PCA), wavelet transforms, and support vector machines (SVMs) [3,4] have been applied to anomaly detection but often suffer from limited generalization capability under dynamic operating regimes or heterogeneous data distributions. More recently, deep learning paradigms—such as long short-term memory (LSTM) networks [5], temporal convolutional networks (TCNs), and Transformer-based encoders [6]—have been explored for forecasting and classification. However, these models often treat sensor channels as independent time series and fail to capture the underlying structural correlations among transformer subsystems.

Therefore, sequence-only models tend to overlook critical cross-component interactions (e.g., thermoelectric couplings between oil, windings, and core), which are essential for accurate state prediction and timely fault localization in power transformers.

To address the spatial structure embedded in sensor networks, graph neural networks (GNNs) have emerged as a promising class of models for learning on structured data [7,8]. In power system applications, GNNs have demonstrated effectiveness in capturing intervariable interactions and improving predictive robustness. For instance, graph convolutional networks (GCNs) have been employed for static graph modeling of sensor layouts [9,10], while attention-based dynamic graphs have been explored for wind turbine health monitoring and grid fault localization [11,12]. Despite these advances, existing GNN-based approaches typically suffer from limitations such as reliance on static or manually designed adjacency matrices, inadequate adaptability to temporal variations, and insufficient incorporation of residual-driven fault reasoning.

In particular, (a) static or heuristic adjacency structures often underfit the evolving dependencies in nonstationary SCADA data; (b) attention-based dynamic graphs improve flexibility but are rarely combined with residual-aware temporal decoders tailored for health evaluation; and (c) most existing models prioritize point forecasting accuracy without offering interpretable indicators suitable for early-warning diagnostics. These gaps—identified through a comprehensive survey of over one hundred transformer fault detection studies involving GNNs and spatiotemporal models—motivate the unified design presented in this work.

In this paper, a residual-aware spatiotemporal graph neural network (STGNN) framework is proposed for transformer health prediction and fault detection. The framework dynamically constructs attention-based sensor graphs at each time step to capture evolving intervariable dependencies. Spatial features are aggregated via Chebyshev spectral graph convolution [13,14], and temporal evolution is modeled using gated recurrent units (GRUs) enhanced with residual connections [15,16]. A health indicator (HI) is computed from node-level prediction residuals [17,18], and early-stage anomalies—such as cooling degradation, insulation aging, and on-load tap changer (OLTC) malfunction—are detected via a quantile-based thresholding mechanism [19,20].

Compared to *static-graph* GCN-based methods, our approach *learns* time-varying dependencies online; relative to attention-based dynamic graph learners, we explicitly integrate

Chebyshev filtering with a residual-enhanced GRU decoder to stabilize and highlight temporal error signals; and unlike sequence-only models (e.g., LSTM, TCN, Transformer), our framework provides a residual-derived HI with quantile thresholds for interpretable anomaly detection aligned with model uncertainty.

The main contributions of this work are summarized as follows:

- Transformer health monitoring is formulated as a dynamic spatiotemporal graph learning task, where time-varying sensor dependencies are captured through attention-based graph construction. This addresses the limitations of static graph models in adapting to dynamic operational contexts.
- A hybrid model architecture is proposed, combining Chebyshev spectral graph convolution and residual-enhanced GRUs to effectively encode spatial interactions and temporal transitions in SCADA data.
- A residual-based health indicator is defined, and a quantile-based thresholding strategy is introduced to enable interpretable and early anomaly detection.
- A synthetic SCADA dataset simulating five transformer subsystems and diverse fault types is constructed. The proposed method is validated against strong baselines through comprehensive forecasting and fault detection experiments.

The remainder of this paper is organized as follows. Sect 2 introduces the transformer system structure and describes the construction of observability-driven health state vectors. Sect 3 presents the proposed STGNN framework, including dynamic graph construction, spectral graph convolution, residual-enhanced temporal modeling, and the health indicator design. Sect 4 details the experimental setup, data generation process, and baseline methods. Sect 5 reports the results of forecasting and anomaly detection experiments, along with comparative analyses. Finally, Sect 6 concludes the paper and outlines limitations and future research directions.

## 2 Related work

Beyond traditional time-series forecasting and static graph-based approaches, recent efforts have explored residual-aware, physics-informed, and adversarially robust frameworks for power system monitoring. Residual-based attention Physics-Informed Neural Networks (PINNs) [21] have been developed to capture transformer aging dynamics under spatiotemporal physical constraints, offering improved interpretability in renewable-integrated power grids. From a cybersecurity perspective, the EVADE framework [22] introduces targeted adversarial false data injection (FDI) attacks to compromise state estimation, while methods such as LESSON [23] and joint adversarial FDI detection models [24] apply deep learning to detect complex attacks on SCADA measurements.

While these methods contribute to enhancing safety and domain fidelity, they are primarily focused on classification under adversarial scenarios or require prior knowledge of physical mechanisms and labeled attack data. In contrast, the objective of this study is to enable robust and interpretable anomaly detection in naturally nonstationary SCADA environments without relying on labeled faults or predefined physics-based models. By integrating dynamic graph learning, residual-informed temporal modeling, and quantile-based health indicators, the proposed framework offers a data-driven and scalable solution complementary to these existing approaches, emphasizing generalizability and early fault interpretability in practical deployments.

This work provides a generalizable, interpretable, and scalable solution for intelligent transformer condition monitoring, and holds strong potential for practical deployment in real-world substation SCADA environments and smart grid infrastructure.

# 3 Transformer health state modeling and graph structure construction

## 3.1 Transformer system architecture and state variable selection

Power transformers are essential components in modern power transmission and distribution systems. Their operational integrity directly affects grid reliability and stability. Due to complex internal structures, prolonged exposure to high voltages, and challenging environmental conditions, transformers are prone to various faults, including winding overheating, core saturation, insulation degradation, and contact wear. Therefore, prior to predictive modeling using graph neural networks, it is critical to systematically analyze the internal architecture and select a set of high-relevance, multi-dimensional state variables as model inputs.

Based on structural and functional decomposition, the transformer is divided into five major subsystems, each associated with distinct fault mechanisms and monitored indicators:

1. **Winding System**: Responsible for electromagnetic energy conversion and susceptible to issues such as local overheating and inter-turn short circuits. Representative variables include winding temperature ($T_{\text{coil}}$), load current ($I_L$), short-circuit impedance, and voltage total harmonic distortion.
2. **Core System**: Provides magnetic flux pathways and is affected by core losses and electromagnetic interference. Key indicators include core temperature ($T_{\text{core}}$), excitation current ($I_{\text{exc}}$), and eddy current loss metrics.
3. **Insulation System**: Comprising solid (e.g., pressboard) and liquid (e.g., transformer oil) insulation, this subsystem governs dielectric strength and thermal stability. Typical variables include partial discharge magnitude ($Q_{\text{pd}}$), moisture content in oil ($H_2O$), and capacitive current ($I_C$).
4. **Cooling System**: Maintains thermal equilibrium through forced oil circulation or air-cooling mechanisms. Monitored variables include oil temperature ($T_{\text{oil}}$), fan or pump operational status, and the temperature differential at the cooling outlet.
5. **Tap Changer System**: Regulates transformer output voltage and exhibits high switching frequency and failure susceptibility. Relevant indicators include tap position ($P_{\text{tap}}$), switching frequency, contact resistance, and transition duration.

Based on these subsystem characteristics, a multi-dimensional raw state vector is constructed as:

$$X_t = \left[ x_1^t, x_2^t, \ldots, x_n^t \right] \in \mathbb{R}^n, \tag{1}$$

where $x_i^t$ denotes the $i$-th observed state variable at time $t$. The total number of monitored variables $n$ typically ranges from 15 to 25, encompassing multimodal information such as temperature, current, voltage, vibration, and acoustic signals. In addition, certain high-level indicators (e.g., thermal equilibrium index, load ratio) are derived through sensor fusion.

These state variables serve as node attributes for the subsequent spatiotemporal graph construction, providing the input features for the graph neural network to capture spatial dependencies among sensor variables and their temporal dynamics. Ultimately, this supports accurate and interpretable health status prediction for transformer subsystems.

### 3.2 Observability-driven construction of health state vectors

In practical transformer monitoring systems, internal health states cannot be directly measured. Instead, these latent states must be inferred from a limited set of externally observable variables. To ensure that the constructed state vectors are both informative and interpretable, we adopt an observability-driven strategy for deriving the state vector $X_t$.

We assume the existence of an implicit mapping from observable variables $Z_t = [z_1^t, z_2^t, ..., z_m^t]^\top \in \mathbb{R}^m$ to the latent health state vector $X_t \in \mathbb{R}^n$, expressed as:

$$X_t = f(Z_t, \theta), \tag{2}$$

where $f(\cdot)$ may be a physics-informed transformation, a statistical regression model, or a neural network-based approximator, and $\theta$ represents auxiliary model parameters.

To improve the observability and compactness of the derived state representation, we incorporate the following three mechanisms:

**1) Physics-constrained modeling:** Domain knowledge in thermal dynamics and electromagnetic behavior is utilized to impose physical consistency and expand the representational capacity of the latent state space.

**2) Feature selection:** Only those observable variables exhibiting high mutual information or structural relevance to target health states are retained, mitigating the risk of overparameterization and redundancy.

**3) Dimensionality reduction:** Principal component analysis (PCA) is applied to high-dimensional measurements to extract dominant components that encapsulate key health-related patterns:

$$\tilde{X}_t = W_{\mathrm{pca}} \cdot Z_t, \tag{3}$$

where $W_{\mathrm{pca}} \in \mathbb{R}^{n \times m}$ is the learned PCA projection matrix. The resulting reduced-dimension vector $\tilde{X}_t$ preserves essential information while enhancing the numerical stability and generalization ability of downstream graph-based modeling.

This observability-aware construction ensures that the spatiotemporal learning process is grounded in both physical interpretability and data-driven relevance, thereby improving the robustness of health state inference in real-world SCADA environments.

### 3.3 Characteristics of SCADA monitoring data

The predictive modeling of transformer operational states is primarily driven by time-series data collected through Supervisory Control and Data Acquisition (SCADA) systems. These systems integrate diverse sensors distributed across the transformer and its auxiliary components to monitor critical variables—such as temperature, current, voltage, power, and switching states—in real time. While SCADA data offer rich information for analysis, they also exhibit several distinct characteristics that introduce challenges for graph-based spatiotemporal modeling:

1. **Multivariate Heterogeneity:** SCADA datasets encompass a wide range of variable types, including continuous signals (e.g., oil temperature, winding temperature, load current) and discrete or binary indicators (e.g., tap changer position, cooling fan status). These variables differ in physical units, semantic meaning, and sampling frequencies. Therefore, preprocessing steps such as normalization, temporal resampling, and one-hot encoding must be performed to ensure compatibility within a unified learning framework.

2. **Temporal Dependency:** The monitored variables exhibit diverse temporal patterns, including long-term trends, periodic cycles, and high-frequency fluctuations. Moreover, time-lagged dependencies are frequently observed (e.g., the delayed response of oil temperature to rapid load increases), which necessitate sequence modeling across multiple temporal horizons.

3. **Spatial Coupling Structure:** Sensor readings are inherently linked to specific physical subsystems of the transformer, resulting in structural couplings among variables. For example, thermal interactions couple the winding, oil, and core temperatures, while electrical relationships exist between load current and terminal voltage. These dependencies should be explicitly captured through graph-based structures that define meaningful edges among sensor nodes.

4. **Missing and Noisy Observations:** Real-world SCADA data are often affected by missing entries (e.g., due to sensor outages or communication delays) and noise corruption. This calls for the use of imputation techniques, signal denoising, and robust learning architectures to mitigate data imperfections and ensure stable predictions.

5. **Operational Diversity and Non-Stationarity:** Transformer operating conditions change over time due to seasonal variations, fluctuating load demands, and grid reconfiguration policies. These dynamics introduce distributional shifts in the data, requiring predictive models to maintain adaptability and generalization capabilities under non-stationary regimes.

In summary, SCADA data represent a complex form of multivariate, heterogeneous, and dynamic industrial time series. To ensure robust and accurate modeling under practical conditions, these characteristics must be systematically integrated into the design of graph neural networks, including input feature selection, dynamic graph construction, and spatiotemporal representation learning.

## 3.4 Graph construction methodology

To capture the intrinsic correlations among transformer operating variables, a dynamic graph representation is constructed from SCADA data, in which each node corresponds to a sensor-derived variable and each edge quantifies the relationship strength between variable pairs. This graph serves as the structural foundation for the proposed STGNN model.

Formally, at each time step $t$, the system is represented as a graph $\mathcal{G}_t = (\mathcal{V}, \mathcal{E}_t, \mathbf{A}_t)$, where $\mathcal{V}$ denotes the node set comprising $n$ selected sensor variables, $\mathcal{E}_t$ is the edge set, and $\mathbf{A}_t \in \mathbb{R}^{n \times n}$ is the weighted adjacency matrix that governs message propagation across nodes.

Each node encodes a specific SCADA variable, such as winding temperature, core temperature, or load current. The associated node features are derived from the health state vector $X_t$, as defined in Sect 3.1.

**Static Edge Construction.** To incorporate long-term structural dependencies, static edge weights are first computed using Pearson correlation coefficients over historical data:

$$\left[\mathbf{A}_{\text{static}}\right]_{ij} = \begin{cases} |\rho_{ij}|, & \text{if } |\rho_{ij}| > \delta, \\ 0, & \text{otherwise,} \end{cases} \tag{4}$$

where $\rho_{ij}$ is the Pearson coefficient between variables $x_i$ and $x_j$, and $\delta$ is a sparsity threshold to retain only significant correlations.

**Dynamic Attention-Based Edges.** To account for nonstationary and nonlinear relationships, an attention mechanism is employed to dynamically compute edge weights at each time

step:

$$[\mathbf{A}_t]_{ij} = \frac{\exp\left(\phi(\mathbf{x}_i^t, \mathbf{x}_j^t)\right)}{\sum_{k=1}^{n} \exp\left(\phi(\mathbf{x}_i^t, \mathbf{x}_k^t)\right)}, \tag{5}$$

with scoring function:

$$\phi(\mathbf{x}_i, \mathbf{x}_j) = \mathbf{a}^\top \sigma\left(\mathbf{W}[\mathbf{x}_i \parallel \mathbf{x}_j]\right), \tag{6}$$

where $\mathbf{x}_i^t, \mathbf{x}_j^t \in \mathbb{R}^d$ are node features, $\mathbf{W} \in \mathbb{R}^{d' \times 2d}$ and $\mathbf{a} \in \mathbb{R}^{d'}$ are trainable parameters, $\parallel$ denotes feature concatenation, and $\sigma(\cdot)$ is a nonlinear activation function (e.g., LeakyReLU).

**Similarity-Based Gaussian Kernel.** To further model feature similarity, a Gaussian kernel is adopted to generate edge weights based on Euclidean distances:

$$[\mathbf{A}_{\text{gauss}}]_{ij} = \exp\left(-\frac{\|\mathbf{x}_i^t - \mathbf{x}_j^t\|^2}{2\sigma^2}\right), \tag{7}$$

where $\sigma$ is a bandwidth parameter controlling sensitivity to local feature variations.

Hybrid Adjacency Aggregation. The final time-varying adjacency matrix is constructed via a convex combination of the above three components:

$$\mathbf{A}_t = \lambda_1 \cdot \mathbf{A}_{\text{static}} + \lambda_2 \cdot \mathbf{A}_{\text{gauss}} + \lambda_3 \cdot \mathbf{A}_{\text{att}}(t), \tag{8}$$

subject to $\lambda_1 + \lambda_2 + \lambda_3 = 1$. These weights are either manually tuned or optimized through validation. This hybrid design enables the model to simultaneously capture long-term structural correlations, real-time feature similarity, and adaptive temporal dynamics, thereby providing a comprehensive input representation for downstream spatiotemporal learning.

## 4 Design of spatiotemporal graph neural network for health status prediction

### 4.1 Model architecture design

To effectively capture both the spatial dependencies and temporal dynamics present in transformer SCADA data, we propose a hybrid modeling framework that integrates graph neural networks (GNNs) with sequence modeling architectures. The goal is to exploit the topological structure among sensor state variables via graph learning, while concurrently modeling their temporal evolution for accurate health status prediction. Our design choices are motivated by prior evidence in related domains: attention-based dynamic graph construction has been shown to improve adaptability to evolving dependencies in nonstationary sensor networks [11,12,18]; Chebyshev spectral graph convolution [13,14] provides extended receptive fields and computational efficiency for graph-structured industrial data; and residual-enhanced GRUs [15,16] improve temporal stability and sensitivity to subtle system drifts. By integrating these components, our STGNN addresses three common limitations in existing works: (1) static-graph models cannot adapt to changing dependencies, (2) sequence-only models neglect cross-variable structure, and (3) dynamic-graph methods without residual-aware decoders lack interpretable, error-driven health indicators.

The overall framework consists of three main components:

1. **Graph Representation Learning Module**, which extracts structural features from a series of dynamically constructed graphs $\mathcal{G}_t = (\mathcal{V}, \mathcal{E}_t, \mathbf{A}_t)$ at each time step.

2. **Temporal Modeling Module**, which captures sequential dependencies across time from node-level embeddings, implemented using Gated Recurrent Units (GRUs).
3. **Health Indicator Definition**, which derives interpretable indicators based on prediction residuals, and enables anomaly detection using threshold-based methods.

Given a sequence of $L$ historical health state vectors $\{\boldsymbol{X}_{t-L+1}, ..., \boldsymbol{X}_t\}$, and their corresponding graphs, the model input is formatted as a spatiotemporal tensor:

$$\mathcal{X} = [\mathbf{X}_{t-L+1}; \mathbf{X}_{t-L+2}; ...; \mathbf{X}_t] \in \mathbb{R}^{L \times n \times d}, \tag{9}$$

where $n$ is the number of monitored variables (nodes), and $d$ denotes the feature dimension per node. For each time step $\tau$, a graph convolutional operation is applied to extract spatial features:

$$\mathbf{H}_\tau = \text{GNN}(\mathbf{X}_\tau, \mathbf{A}_\tau), \quad \tau = t - L + 1, ..., t. \tag{10}$$

In our implementation, the GNN layer is instantiated as a Chebyshev spectral graph convolution [13,14], allowing efficient localized filtering over the graph using polynomial approximation of the Laplacian:

$$\mathbf{H}_\tau = \sum_{k=0}^{K} \theta_k T_k(\tilde{\mathbf{L}}_\tau) \mathbf{X}_\tau, \tag{11}$$

where $T_k$ is the $k$-th order Chebyshev polynomial, and $\tilde{\mathbf{L}}_\tau$ is the scaled Laplacian of $\mathbf{A}_\tau$. We set $K = 3$ to balance receptive field size and efficiency.

The sequence of graph embeddings is then passed through a GRU network to learn temporal dependencies:

$$\mathbf{h}_t^{\text{pred}} = \text{GRU}(\mathbf{H}_{t-L+1}, ..., \mathbf{H}_t), \tag{12}$$

To improve temporal stability, the GRU module is augmented with a residual connection:

$$\mathbf{h}_t^{\text{res}} = \text{GRU}(\mathbf{H}_t, \mathbf{h}_{t-1}) + \mathbf{H}_t. \tag{13}$$

This structure helps preserve low-amplitude but persistent residual deviations, which often signal early-stage faults.

Mapped to the predicted state estimate for the next time step using a fully connected projection:

$$\hat{\mathbf{X}}_{t+1} = \text{FC}(\mathbf{h}_t^{\text{res}}). \tag{14}$$

Based on the predicted and observed states, a root mean square error (RMSE)-based health indicator (HI) is computed to quantify residual deviations from expected behavior.

The complete architecture is illustrated in Fig 1, which delineates the three core modules—graph-based representation learning, temporal sequence modeling, and health indicator-based anomaly detection—and their interactions.

## 4.2 Graph representation learning module

To model the complex and time-varying dependencies among transformer state variables, we incorporate a self-attention mechanism into the graph neural network to dynamically construct the graph topology. This enables adaptive learning of spatial relationships and facilitates effective information propagation via graph convolution.

Graph Representation          Temporal            Health Indicator
Learning Module        Modeling Module            Definition

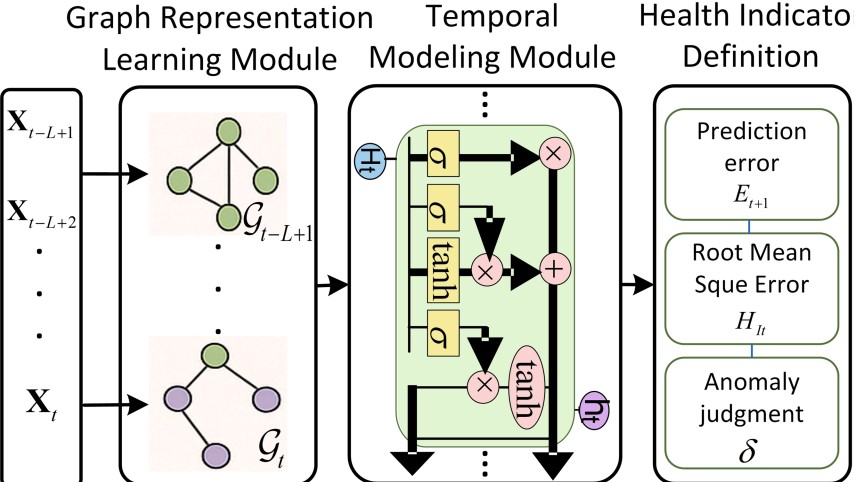

**Fig 1. TOverall model architecture combining graph representation learning, temporal modeling via GRU, and health indicator computation for anomaly detection.**

At each time step $t$, the system is represented by a multivariate input vector $X_t = [x_1^t, ..., x_n^t]^\top \in \mathbb{R}^n$, where each node corresponds to a monitored sensor variable. Each node feature $\mathbf{x}_i^t \in \mathbb{R}^d$ represents the temporal state of the $i$-th sensor at time $t$, derived from a sliding window of recent measurements.

To capture adaptive spatial relationships, a dynamic graph is constructed using a pairwise attention mechanism. Inspired by TempGNN [12], the attention score between any pair of nodes $i$ and $j$ is computed as:

$$e_{ij}^t = \mathbf{a}^\top \sigma \left( \mathbf{W} \cdot [\mathbf{x}_i^t \,\|\, \mathbf{x}_j^t] \right), \tag{15}$$

where $\mathbf{x}_i^t, \mathbf{x}_j^t \in \mathbb{R}^d$ are input features, $[\cdot \,\|\, \cdot]$ denotes concatenation, $\mathbf{W} \in \mathbb{R}^{d' \times 2d}$, $\mathbf{a} \in \mathbb{R}^{d'}$ are learnable parameters, and $\sigma(\cdot)$ is a nonlinear activation function. This design allows the graph to adapt to both static and dynamic variable interactions.

The attention scores are normalized to yield the time-varying adjacency matrix $\mathbf{A}_t$:

$$[\mathbf{A}_t]_{ij} = \frac{\exp(e_{ij}^t)}{\sum_{k=1}^{n} \exp(e_{ik}^t)}. \tag{16}$$

The resulting $\mathbf{A}_t$ encodes a soft, directed graph structure that evolves with time, allowing the model to emphasize relevant sensor interactions under varying operating conditions.

With the constructed graph, node embeddings are computed using spectral graph convolution. Specifically, we adopt the Chebyshev polynomial-based graph convolution (ChebNet), defined as:

$$\mathbf{H}_t = \sum_{k=0}^{K} \theta_k \cdot T_k(\tilde{\mathbf{L}}) \cdot \mathbf{X}_t, \tag{17}$$

where $T_k(\cdot)$ denotes the Chebyshev polynomial of order $k$, $\tilde{\mathbf{L}} = \frac{2}{\lambda_{\max}}\mathbf{L} - \mathbf{I}$ is the scaled Laplacian matrix, and $\theta_k$ are trainable spectral coefficients.

Compared to traditional GCNs that use truncated first-order filtering, the ChebNet allows a broader receptive field and faster convergence, especially when $K > 1$. In this work, $K = 3$ is selected to balance model expressiveness and computational cost.

To validate the robustness of the graph modeling strategy, we additionally implement two variants for comparison: (1) static GCN using a fixed adjacency matrix based on correlation similarity, and (2) Graph Attention Network (GAT) with multi-head attention. These are used in ablation experiments to assess the benefits of dynamic attention graphs in STGNN.

In summary, this module integrates dynamic attention-based graph construction and localized Chebyshev spectral filtering, enabling the model to learn context-aware spatial dependencies across transformer subsystems in a data-driven and interpretable manner.

## 4.3 Temporal modeling module

To capture the temporal evolution of multivariate transformer state variables, we employ a Gated Recurrent Unit (GRU) architecture to process the graph-based node embeddings. GRUs are known for their parameter efficiency and robustness against vanishing gradients, making them suitable for modeling short- and mid-term dependencies in industrial time series.

Let the sequence of graph node embeddings be denoted as:

$$\mathcal{H} = \left[ \mathbf{H}_{t-L+1}, \mathbf{H}_{t-L+2}, \ldots, \mathbf{H}_t \right] \in \mathbb{R}^{L \times n \times d}, \tag{18}$$

where $L$ is the input window length, $n$ is the number of nodes, and $d$ is the embedding dimension.

The GRU updates at each time step $\tau$ are defined as:

$$
\begin{aligned}
\mathbf{z}_\tau &= \sigma\left(\mathbf{W}_z \mathbf{H}_\tau + \mathbf{U}_z \mathbf{h}_{\tau-1}\right), \\
\mathbf{r}_\tau &= \sigma\left(\mathbf{W}_r \mathbf{H}_\tau + \mathbf{U}_r \mathbf{h}_{\tau-1}\right), \\
\tilde{\mathbf{h}}_\tau &= \tanh\left(\mathbf{W}_h \mathbf{H}_\tau + \mathbf{U}_h \left(\mathbf{r}_\tau \odot \mathbf{h}_{\tau-1}\right)\right), \\
\mathbf{h}_\tau &= \left(1 - \mathbf{z}_\tau\right) \odot \mathbf{h}_{\tau-1} + \mathbf{z}_\tau \odot \tilde{\mathbf{h}}_\tau
\end{aligned}
\tag{19}
$$

where, $\sigma(\cdot)$ denotes the sigmoid activation, and $\odot$ represents element-wise multiplication.

The use of GRUs allows the model to retain temporal dependencies while avoiding over-fitting or instability from long-range correlations. In our model, the GRU operates on node-level graph embeddings and produces a temporally smoothed hidden state sequence.

To enhance anomaly sensitivity and temporal generalization, we incorporate a residual decoding strategy. Specifically, the final hidden state $\mathbf{h}_t^{\text{pred}}$ is used to regress a residual prediction offset:

$$\hat{\mathbf{X}}_{t+1} = \mathbf{X}_t + \mathbf{R}_{t+1}, \tag{20}$$

where the residual term is computed as:

$$\mathbf{R}_{t+1} = \text{MLP}\left(\mathbf{h}_t^{\text{pred}}\right), \tag{21}$$

and MLP denotes a fully connected multilayer perceptron.

This residual formulation not only improves convergence and stability of the temporal decoder but also facilitates downstream fault detection by explicitly linking prediction errors to abnormal behavior.

In addition, a node-wise residual weighting mechanism is used during training to highlight critical variables in health state forecasting:

$$\mathcal{L} = \sum_{i=1}^{n} \alpha_i \cdot \left( x_i^{t+1} - \hat{x}_i^{t+1} \right)^2, \tag{22}$$

where $\alpha_i$ is the contribution weight of node $i$, determined either via domain knowledge or learned during training.

This weighted loss encourages the model to prioritize key indicators such as winding temperature, core flux, or partial discharge amplitude, which are highly correlated with early transformer degradation.

In summary, the temporal modeling module combines GRU-based sequence encoding, residual-based prediction decoding, and node-aware loss weighting to provide a robust and interpretable framework for time-dependent health state estimation.

## 4.4 Health indicator definition and anomaly detection mechanism

To facilitate quantifiable evaluation of transformer operational status and enable early fault detection, we define a health indicator (HI) based on prediction residuals. This indicator reflects the model's confidence in its estimation and serves as a proxy for system degradation or abrupt anomalies.

Given the predicted state $\hat{\mathbf{X}}_{t+1} \in \mathbb{R}^n$ and the ground-truth observation $\mathbf{X}_{t+1} \in \mathbb{R}^n$, the residual vector is defined as:

$$\mathbf{E}_{t+1} = \mathbf{X}_{t+1} - \hat{\mathbf{X}}_{t+1}. \tag{23}$$

We compute the root mean square error (RMSE) over all nodes as the global health indicator:

$$\mathrm{HI}_{t+1} = \sqrt{\frac{1}{n} \sum_{i=1}^{n} \left( x_i^{t+1} - \hat{x}_i^{t+1} \right)^2}, \tag{24}$$

A larger HI value indicates greater deviation from normal behavior, signaling potential degradation or operational abnormality.

To identify anomalous time steps, we introduce a detection threshold $\delta$ based on the empirical distribution of HI values on a validation dataset. Specifically, the threshold is chosen as the 95th percentile:

$$\delta = \mathrm{Quantile}_{0.95}\left( \{ \mathrm{HI}_\tau \}_{\tau \in \mathrm{val}} \right). \tag{25}$$

If $\mathrm{HI}_{t+1} > \delta$, the corresponding time step is flagged as abnormal.

To reduce false positives caused by transient fluctuations, we apply a post-processing rule based on temporal continuity. An anomaly is confirmed only if HI exceeds $\delta$ for at least $k$ consecutive time steps:

$$\mathrm{Abnormal}(t+1) = \begin{cases} 1, & \text{if } \mathrm{HI}_{t+1}, \dots, \mathrm{HI}_{t+k} > \delta, \\ 0, & \text{otherwise.} \end{cases} \tag{26}$$

This indicator-based mechanism offers high interpretability, low computational cost, and easy integration into industrial alarm systems and maintenance workflows.

# 5 Experiment and result analysis

## 5.1 Experimental setup and data description

To evaluate the effectiveness of the proposed STGNN in modeling transformer health states, we construct a simulation-based SCADA dataset grounded in real-world operational characteristics of a 110 kV oil-immersed power transformer.

A total of $n = 18$ sensor variables are monitored, covering the five major subsystems of the transformer:

- **Winding system**: high-voltage and low-voltage winding temperature, load current;
- **Core system**: core temperature, magnetic flux density;
- **Insulation system**: partial discharge magnitude, dielectric loss factor;
- **Cooling system**: oil temperature, cooler fan status, oil flow rate;
- **Tap changer (OLTC)**: tap position, motor operating current.

Additionally, ambient temperature and oil level are included to reflect environmental influence and volume compensation effects. All variables are sampled every 1 minute over 30 consecutive days, yielding $T \approx 43{,}200$ time steps.

Multiple fault types are embedded into the data to simulate realistic abnormal behavior, including:

- *Cooling system failure*—simulated by a gradual rise in oil and winding temperatures with stagnant cooler fan status;
- *Tap changer malfunction*—characterized by erratic changes in tap position and spike in motor current;
- *Insulation degradation*—reflected by an increase in partial discharge and dielectric loss over time.

Model inputs are constructed using a sliding window of length $L = 12$, capturing the past 12 time steps to predict the state vector at time $t + 1$. The resulting input tensor has dimensions $\mathbb{R}^{12 \times 18 \times 1}$, with each node representing a standardized scalar reading of a sensor channel.

All variables are normalized using Z-score standardization. Missing values, if any, are linearly interpolated. Ground-truth anomaly labels are reserved exclusively for evaluation purposes and are not used during model training.

All experiments were implemented in Python 3.10 using PyTorch 2.1 and the Deep Graph Library (DGL) 1.1. The model was trained on a workstation equipped with an NVIDIA RTX 3090 GPU (24GB), AMD Ryzen 9 5950X CPU, and 64GB RAM. Random seeds were fixed across all runs to ensure experimental reproducibility.

The model is trained on an NVIDIA RTX 3090 GPU with the Adam optimizer (initial learning rate 0.001), batch size of 64, and up to 200 training epochs. Early stopping is employed based on validation loss. The loss function is the node-weighted mean squared error as described in Sect 3.3.

For graph construction, the adjacency matrix at each time step combines static Pearson correlation, Gaussian kernel similarity, and attention-derived dynamic edges using a convex combination:

$$\mathbf{A}_t = \lambda_1 \mathbf{A}_{\text{static}} + \lambda_2 \mathbf{A}_{\text{gauss}} + \lambda_3 \mathbf{A}_{\text{att}}(t), \tag{27}$$

where $\lambda_1 = 0.3$, $\lambda_2 = 0.3$, and $\lambda_3 = 0.4$. These weights were selected based on validation performance.

Graph operations are implemented in PyTorch with the Deep Graph Library (DGL). For each time step, the graph structure is dynamically constructed using attention-enhanced adjacency matrices. The spatial modeling employs Chebyshev graph convolutions with polynomial order $K = 3$, facilitating localized topological filtering.

Evaluation metrics for forecasting include Root Mean Square Error (RMSE), Mean Absolute Percentage Error (MAPE), and Maximum Absolute Error. For anomaly detection, we compute Accuracy, Precision, Recall, and F1-score using binary classification of health indicator thresholds. All metrics follow standard definitions and are reported over both clean and noisy test intervals.

This setup reflects the real-world sensing architecture of transformer substations, and supports an end-to-end learnable pipeline for spatiotemporal modeling, enabling future extension to online health monitoring under IEC 60076-7 framework.

## 5.2 Evaluation metrics and baseline methods

To rigorously assess the proposed STGNN model in the context of transformer state forecasting and fault detection, we establish a comprehensive evaluation framework that captures both regression accuracy and classification reliability. The objective is to quantify the model's ability to anticipate operating deviations and accurately identify potential faults based on learned spatiotemporal representations.

For the forecasting task, we employ Root Mean Square Error (RMSE) and Mean Absolute Percentage Error (MAPE) as the primary regression metrics. RMSE reflects the overall prediction deviation magnitude and is defined as:

$$\text{RMSE} = \sqrt{\frac{1}{n} \sum_{i=1}^{n} \left( x_i^{t+1} - \hat{x}_i^{t+1} \right)^2}, \tag{28}$$

MAPE quantifies relative deviations with respect to the true state values and is formulated as:

$$\text{MAPE} = \frac{100\%}{n} \sum_{i=1}^{n} \left| \frac{x_i^{t+1} - \hat{x}_i^{t+1}}{x_i^{t+1} + \epsilon} \right|, \quad \epsilon = 10^{-6}, \tag{29}$$

We additionally report the maximum node-level absolute error (Max-Error) to capture worst-case prediction discrepancies across all monitored transformer variables, such as winding temperature or partial discharge amplitude.

For anomaly detection evaluation, the task is cast as a binary classification problem using the health indicator (HI) derived in Sect 3.4. We adopt four standard metrics computed from the confusion matrix: Accuracy, Precision, Recall, and F1-score. These metrics reflect the model's capability to correctly detect emerging abnormal conditions while minimizing false alarms.

To benchmark our method, we compare STGNN with four representative baselines featuring distinct spatial-temporal modeling paradigms:

- **LSTM:** pure sequence modeling without graph structure;
- **GCN-LSTM:** static graph convolution followed by temporal encoding;
- **StemGNN:** spectral temporal graph modeling using SVD and wavelets;
- **TempGNN:** dynamic attention-based graph generation with GRU modeling.

Our proposed STGNN integrates Chebyshev spectral graph convolution and dynamic graph attention, jointly optimized with residual-enhanced temporal modeling.

Table 1 summarizes the architectural components of all compared models.

## 5.3 Model performance evaluation

To evaluate the effectiveness of the proposed STGNN in multivariate transformer state forecasting, we conduct quantitative comparisons against representative baseline models under identical experimental settings. All methods are trained and tested using the same input sequences, loss functions, and optimization hyperparameters to ensure fairness.

Table 2 reports the prediction results in terms of Root Mean Square Error (RMSE), Mean Absolute Percentage Error (MAPE), and maximum node-level residuals. The proposed STGNN outperforms all baselines across all metrics. Compared to LSTM and GCN-LSTM, STGNN achieves substantial improvements in both absolute error reduction and relative percentage accuracy. This indicates the advantage of jointly modeling dynamic graph structures and residual-enhanced temporal dependencies, especially for multivariate SCADA signals with nonstationary behaviors. In particular, the lowest Max Error value reflects the model's stability in forecasting high-impact signals such as winding temperature and partial discharge, which are critical for early-stage degradation detection.

Fig 2 shows the RMSE convergence curves across 50 training epochs. STGNN achieves the lowest plateau RMSE and converges faster than the baselines, benefiting from the spatiotemporal coupling of graph attention and residual-enhanced temporal encoding.

Notably, the convergence curve of STGNN is smoother and more stable than those of GCN-LSTM and StemGNN, which exhibit larger oscillations throughout training. This suggests that the residual-enhanced GRU module effectively stabilizes gradient updates during backpropagation, thereby accelerating convergence and reducing the risk of overfitting or premature plateaus. In addition, the dynamic graph attention mechanism allows the model to flexibly adapt to evolving inter-sensor dependencies at each time step, which contributes to better generalization and predictive consistency.

**Table 1. Architectural components of baseline methods and the proposed STGNN.**

| Method | Spatial Module | Temporal Module | Feature Strategy |
|---|---|---|---|
| LSTM | None | LSTM | Raw variable sequence |
| GCN-LSTM | Static GCN | LSTM | Graph-enhanced node embedding |
| StemGNN | Spectral decomposition | Wavelet transform | Dynamic spectral input |
| TempGNN | Attention-based dynamic graph | GRU | Residual-enhanced input |
| **Ours (STGNN)** | ChebNet + dynamic attention | GRU + residual modeling | Multi-view graph + node weighting |

Table 1 summarizes the spatial, temporal, and feature modeling components of all baseline methods, emphasizing how each architecture encodes transformer monitoring signals.

**Table 2. Prediction performance of different models on transformer state forecasting.**

| Method | RMSE | MAPE (%) | Max Error |
|---|---|---|---|
| LSTM | 0.215 | 11.2 | 0.512 |
| GCN-LSTM | 0.192 | 9.8 | 0.447 |
| StemGNN | 0.181 | 9.2 | 0.421 |
| TempGNN | 0.176 | 8.9 | 0.392 |
| **Ours (STGNN)** | **0.149** | **7.3** | **0.336** |

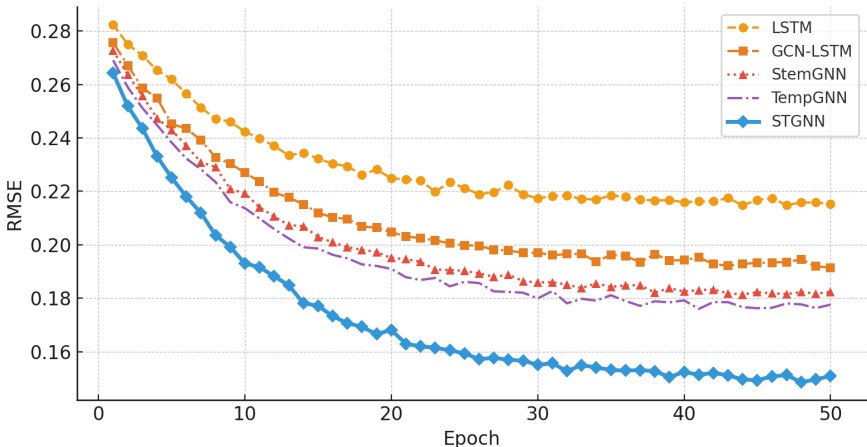

**Fig 2. The elastic deflection of the flexible spacecraft with PD control.**

From a practical standpoint, fast and stable convergence is particularly beneficial in online or incremental training scenarios, where the model must rapidly adapt to new SCADA data without extended retraining. These results demonstrate that the architectural design of STGNN not only improves final forecasting accuracy, but also enhances optimization behavior during the training process.

To further assess generalization under different operational conditions, we partition the test set into noise-free and noise-contaminated intervals. As shown in Fig 3, STGNN maintains superior anomaly detection performance in both regimes. It yields higher F1-scores in noisy segments, suggesting enhanced robustness to sensor noise and dynamic disturbances.

In summary, STGNN demonstrates consistent advantages in prediction accuracy, training efficiency, and fault detection reliability, underscoring its practical applicability for transformer health monitoring in SCADA environments.

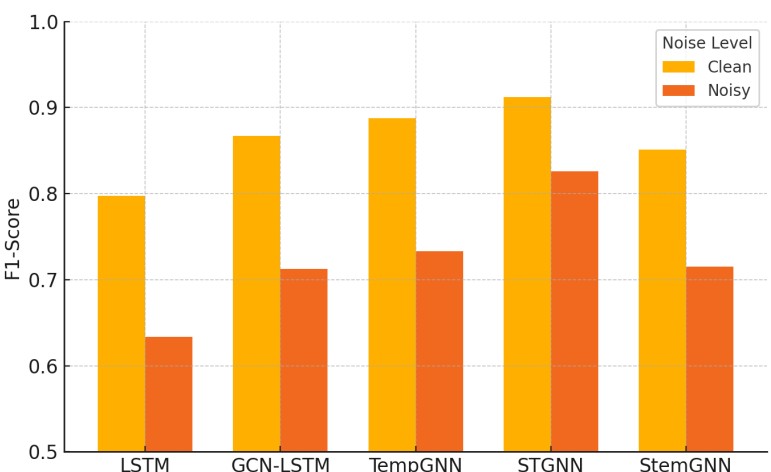

**Fig 3. F1-score comparison of five models under clean and noisy intervals.**

## 5.4 Health indicator and anomaly detection analysis

Based on the transformer state forecasting results, we construct a Health Indicator (HI) derived from RMSE to quantify residual deviations from normal operating behavior. A quantile-based thresholding strategy is then applied for anomaly identification. This section assesses the proposed HI framework in terms of fault detection accuracy, sensitivity to early degradation, and robustness against noise.

Fig 4 presents a representative HI sequence from the test set, comparing the outputs of different models against the annotated ground truth fault interval. The STGNN model exhibits a distinct upward deviation in HI before the onset of failure, enabling early fault warning. This sharp rise reflects the model's sensitivity to subtle degradation patterns, which accumulate gradually before becoming critical. By contrast, LSTM and GCN-LSTM show flatter or delayed changes in their HI curves, indicating reduced responsiveness to weak early-stage deviations.

Moreover, the temporal alignment of HI spikes from STGNN closely matches the onset of true failure intervals, enabling better fault localization in time. This precise timing is essential for real-world SCADA systems where maintenance scheduling or alert escalation must be both early and trustworthy.

To quantify detection performance, we compute standard classification metrics: Accuracy, Precision, Recall, and F1-score, over the entire test set. As reported in Fig 3, STGNN achieves F1-scores of 0.91 and 0.82 under clean and noisy conditions respectively, outperforming all baselines. Importantly, the recall metric of STGNN remains high even under noise injection, while other models (especially LSTM) suffer significant drops. This suggests that the residual-based HI—when paired with dynamic attention graphs—is more robust to noise, as it emphasizes structural deviations rather than isolated signal fluctuations.

To evaluate the stability of the HI thresholding scheme, we conduct a sensitivity analysis across multiple quantile settings ranging from 90% to 99%. Results indicate that the 95th percentile provides the optimal trade-off between false positives and missed detections. We observe that STGNN maintains consistent F1-scores across different thresholds, while LSTM and GCN-LSTM exhibit greater performance fluctuations. This reflects the stability of the residual signal distribution generated by our model, making it more suitable for integration into alarm-based monitoring workflows.

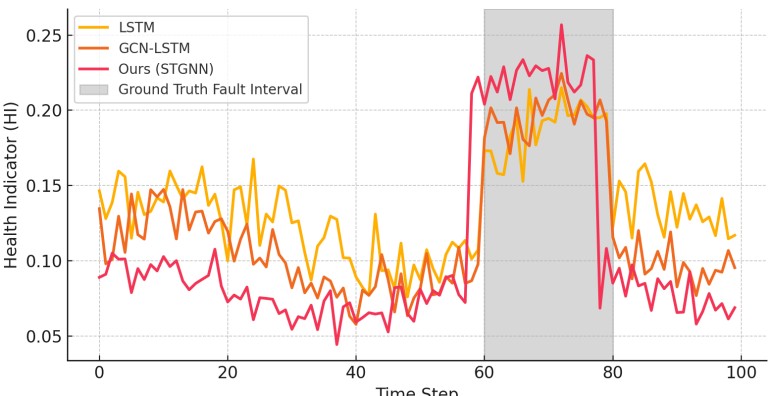

**Fig 4. Comparison of HI curves for different models and the annotated fault interval.** The proposed STGNN demonstrates earlier and sharper deviation, indicating superior fault alerting capability and temporal localization.

In real-world deployments, robustness to noise, stable decision thresholds, and early fault sensitivity are essential features. The results in this section demonstrate that the proposed STGNN design satisfies these requirements through its combination of temporal residual modeling and adaptive graph attention.

In summary, the STGNN framework—by explicitly encoding residual trends and topological dependencies—enables early, accurate, and noise-resilient detection of transformer anomalies, making it well-suited for deployment in practical SCADA-based condition monitoring systems.

## 6 Conclusion and future work

### 6.1 Conclusion

In this study, we propose a STGNN framework for transformer health state prediction and anomaly detection, which jointly models dynamic inter-variable relationships and temporal evolution in SCADA data. By integrating ChebNet-based graph convolutions with residual-enhanced GRU forecasting, the model effectively captures topological correlations and time-series patterns across key transformer subsystems. A health indicator (HI) is derived from prediction residuals to support early fault detection via a quantile-based thresholding mechanism. Experimental results on a simulated SCADA dataset reflecting five typical subsystems— winding, core, cooling, insulation, and tap changer—demonstrate that the proposed method achieves superior RMSE and MAPE scores in state forecasting and significantly outperforms baseline models in anomaly identification under both clean and noisy conditions. The proposed STGNN framework thus offers a scalable and interpretable solution for intelligent condition monitoring of transformers. Future work will focus on extending this approach to real-world deployment, integrating multi-source sensing data such as DGA and vibration signals, and incorporating explainable GNN techniques to enhance model transparency and decision support.

### 6.2 Limitations and future work

While the proposed STGNN framework demonstrates strong performance in forecasting and anomaly detection, several limitations remain to be addressed in future work.

- **Synthetic Dataset Generalizability.** All experiments were conducted using a synthetic SCADA dataset designed to emulate realistic transformer behavior. Although care was taken to simulate various fault types and subsystem dynamics, real-world variability may involve unmodeled patterns, non-Gaussian noise, and external disturbances. Validation on real SCADA logs from industrial substations will be pursued in future studies to assess deployment feasibility.
- **Full Observability Assumption.** The current model assumes that all sensor variables are fully observed at each time step. However, in real-world SCADA environments, signal dropout, packet loss, and asynchronous sampling often occur. Future work will explore spatiotemporal imputation strategies and semi-supervised graph learning methods to address missing or sparse observations.
- **Model Interpretability.** Although the proposed residual-based health indicator improves interpretability over pure black-box forecasting models, the internal graph attention mechanisms remain opaque. Further efforts will be made to incorporate graph explainability tools (e.g., GNNExplainer or counterfactual attention tracing) to interpret node-level contributions to fault detection decisions.

These limitations motivate future work toward real-world validation, robustness to incomplete data, and greater transparency in decision-making for power system diagnostics.

## Supporting information

**S1 Text. Paper program.**
(PDF)

**S2 Dataset. Minimal_SCADA_Dataset.** The minimal anonymized dataset used for validating transformer state forecasting and anomaly detection results, including five key subsystems and their associated sensor time series.
(CSV)

## Author contributions

**Conceptualization:** Peng Zeng, Gong Chu.

**Data curation:** Peng Zeng, Gong Chu.

**Formal analysis:** Peng Zeng, Gong Chu.

**Funding acquisition:** Gong Chu.

**Investigation:** Peng Zeng.

**Methodology:** Peng Zeng.

**Project administration:** Peng Zeng.

**Resources:** Peng Zeng.

**Software:** Peng Zeng, Gong Chu.

**Supervision:** Peng Zeng, Gong Chu.

**Validation:** Peng Zeng, Gong Chu.

**Visualization:** Peng Zeng, Gong Chu.

**Writing – original draft:** Peng Zeng, Gong Chu.

**Writing – review & editing:** Peng Zeng, Gong Chu.

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
