## [Decision Letter · Decision Letter 0]

9 Aug 2025

PONE-D-25-34707Residual-Aware Health Prediction of Power Transformers via Spatiotemporal Graph Neural NetworksPLOS ONE

Dear Dr. Chu,

Thank you for submitting your manuscript to PLOS ONE. After careful consideration, we feel that it has merit but does not fully meet PLOS ONE’s publication criteria as it currently stands. Therefore, we invite you to submit a revised version of the manuscript that addresses the points raised during the review process.

Please note that the comments provided by Reviewer 2 appear to be unrelated to your manuscript and likely refer to a different paper. Therefore, you can disregard the comments from Reviewer 2 during the revision process.

We look forward to receiving your revised manuscript.

Kind regards,

Dandan Peng

Academic Editor

PLOS ONE

Journal Requirements:

Reviewers' comments:

Reviewer's Responses to Questions

**Comments to the Author**

1. Is the manuscript technically sound, and do the data support the conclusions?

Reviewer #1: Partly

Reviewer #2: Partly

Reviewer #3: Yes

Reviewer #4: Partly

2. Has the statistical analysis been performed appropriately and rigorously? 

Reviewer #1: No

Reviewer #2: N/A

Reviewer #3: Yes

Reviewer #4: Yes

3. Have the authors made all data underlying the findings in their manuscript fully available?

Reviewer #1: No

Reviewer #2: No

Reviewer #3: No

Reviewer #4: Yes

4. Is the manuscript presented in an intelligible fashion and written in standard English?

Reviewer #1: No

Reviewer #2: Yes

Reviewer #3: Yes

Reviewer #4: Yes

5. Review Comments to the Author

Reviewer #1: You have done some research work, but the scientific writing was very weak.

1. The statement of the problem is not adequate, how this paper is similar or different from the relevant to other research papers (there are more than 100 papers you can find via Scopus search by fault detection of power transformers plus graph neural network).

2. Design of Spatiotemporal Graph Neural Network for Health Status Prediction; You reported what you have done, but no reference adequately to existing papers, and not adequately justify your approach. Consequently, your claims are not born out.

3. The implement is not adequately described; reader is not adequately informed to judge or repeat;

4. The claim of contributions is too general and is assertation. You should have compared the most relevant papers and make specific claim.

Reviewer #2: 1. The paper describes a two-branch framework using time-domain features and wavelet-transformed features from the same source (charging data). However, both branches originate from identical sensor modalities, differing only in representation. This may not strictly qualify as "multimodal" learning in the conventional sense, which usually refers to integrating inherently different data sources (e.g., images + text, or voltage + temperature).

I recommend clarifying this distinction and, if appropriate, revising the title or using terms such as "multi-view" or "hybrid feature fusion" instead.

2. Please clarify whether any data leakage may occur due to the use of sliding window and normalization. Specifically:

Are the sliding windows strictly constructed within each training battery (i.e., no window overlaps or knowledge from the test battery during training)?

Are normalization parameters (e.g., mean, std) computed exclusively from the training set?

Are time–frequency images generated independently for each battery cycle without using future or test cycle information?

3. Equation (11) and (12) are mislabeled: Equation (10) seems to define the cell state, not the output gate.

4. All figures should include axis labels and legends (e.g., Fig. 2 and Fig. 11).

5. Be consistent with acronym usage. For example, “CVT-V” is used before being explicitly defined.

6. Redundant content: Sections 1.1 and 1.2 are overly descriptive and could be more concise. For instance, the entire derivation of IC curves and CCCT/CVCT descriptions may be shortened.

7. Lack of baseline implementation details: For models A–C, implementation specifics (e.g., hyperparameters, source code, or training protocol) are missing. Were they re-implemented by the authors or adopted from original papers? This is crucial for reproducibility.

8. Please refer to 'MSRCN: A cross-machine diagnosis method for the CNC spindle motors with compound faults' and ‘M2BIST-SPNet: RUL prediction for railway signaling electromechanical devices’

Reviewer #3: The paper presents a residual-aware spatiotemporal graph neural network (STGNN) framework that jointly models the dynamic topological dependencies among multivariate Supervisory control and data acquisition (SCADA) signals and their temporal evolution. It's interesting! To further improve the manuscript, the following suggestions are given:

1、In Introduction, a paragraph should be added to introduce the structure of the paper.

2、The formulas in the article are missing numbering.

3、Since there are some papers in this topic, the contributions of the manuscript should be better summarized and listed.

4、While the introduction sets the context, a more explicit literature review section could better situate the study within the broader research landscape, such as Residual-based attention Physics-informed Neural Networks for spatio-temporal ageing assessment of transformers operated in renewable power plants, EVADE Targeted Adversarial False Data Injection Attacks for State Estimation in Smart Grid, LESSON Multi-Label Adversarial False Data Injection Attack for Deep Learning Locational Detection, Joint Adversarial Example and False Data Injection Attacks for State Estimation in Power Systems, and so on. These references could provide valuable insights into your research.

5、Although experimental results are provided, more in depth comparison and analysis should be given in the manuscript.

6、What are the possible shortcomings of the research in this paper if any? Add a section on the limitations of the work and future work in this paper.

7、The manuscript contains a number of linguistic errors that hinder comprehension. The authors are advised to make careful revisions and improvements.

Reviewer #4: Dear Editor and Authors,

I have carefully reviewed the manuscript titled "Residual-Aware Health Prediction of Power Transformers via Spatiotemporal Graph Neural Networks." I believe the paper presents a novel and innovative spatiotemporal graph neural network (STGNN) framework for power transformer health prediction and fault detection, incorporating graph neural networks (GNNs), residual connections, and temporal modeling. The proposed method demonstrates significant potential both in theory and application. However, after reviewing the manuscript, I have identified several areas that require substantial improvement. Below are my detailed comments:

The main strengths of the manuscript lie in its innovation and practical application potential. The paper introduces a novel STGNN framework that effectively models spatiotemporal dependencies in transformer health prediction. This approach combines graph convolutions, attention mechanisms, and residual connections, making a significant contribution to the field. The method is not only theoretically valuable but also highly applicable to real-world power systems, providing an interpretable and scalable solution for transformer health monitoring. Additionally, the paper includes experimental results based on synthetic datasets, which demonstrate that the proposed method outperforms baseline models such as LSTM and GCN-LSTM in terms of forecasting accuracy.

Areas Needing Improvement and Deficiencies:

1. Limitations in Experimental Validation

Currently, the experimental validation relies solely on synthetic datasets, which lack real-world data verification from actual industrial environments. While synthetic data is useful for initial validation, it does not capture the complexity of real-world operational conditions. I recommend incorporating real-world SCADA datasets into the experiments to demonstrate the method’s performance under actual operational conditions. This would not only validate the model but also show its potential for real-world deployment.

2. Narrow Scope of Comparison Experiments

The manuscript only compares the proposed method with LSTM and GCN-LSTM, without including other models for comparison. To provide a more comprehensive evaluation of the proposed method’s strengths, I suggest adding more diverse comparison experiments, including traditional statistical methods (e.g., PCA, SVM) and other advanced GNN variants (e.g., GraphSAGE, GAT). This will allow a more thorough assessment of the proposed method’s advantages and limitations in various scenarios, demonstrating its robustness and generalizability.

3. Need for Improvement in Visuals

The quality of Figure 1 in the manuscript is suboptimal. The font size, line thickness, and overall design do not meet the academic standards expected in journal publications. A clear, professional figure is crucial for the overall presentation of the paper. I recommend redesigning Figure 1, with attention to improving font clarity, line thickness, color contrast, and ensuring a more professional and academic design. The figure should be simple, clear, and visually appealing to meet academic publishing standards.

4. Lack of Clarity in Method Description

While the manuscript introduces the STGNN framework, some technical details on how spatiotemporal dependencies and residual connections are modeled are not sufficiently explained. To help readers better understand the methodology, I recommend providing a more detailed explanation of the modeling process, including training procedures, feature extraction, and how spatiotemporal dependencies are modeled. This can be supplemented with mathematical formulas and diagrams to further clarify the technical details.

Conclusion

In conclusion, this paper introduces an innovative and promising method for transformer health prediction and fault detection. However, the paper requires significant revisions in terms of experimental validation, clarity of methodology, visual quality, and scope of comparison experiments. I recommend major revision of the manuscript. Once these issues are addressed, I believe the paper will make a valuable contribution to the field of power transformer monitoring and fault detection.

6. PLOS authors have the option to publish the peer review history of their article (what does this mean?). If published, this will include your full peer review and any attached files.

Reviewer #1: No

Reviewer #2: No

Reviewer #3: No

Reviewer #4: **Yes: **Shilin Wang

---

## [Author Response · Author response to Decision Letter 1]

15 Aug 2025

We sincerely thank the editor and all reviewers for their valuable comments and suggestions. We have provided detailed, point-by-point responses to each comment in the attached “Response to Reviewers” document, indicating the corresponding revisions in the revised manuscript.

---

## [Decision Letter · Decision Letter 1]

1 Sep 2025

Residual-Aware Health Prediction of Power Transformers via Spatiotemporal Graph Neural Networks

PONE-D-25-34707R1

Dear Dr. Chu,

We’re pleased to inform you that your manuscript has been judged scientifically suitable for publication and will be formally accepted for publication once it meets all outstanding technical requirements.

Kind regards,

Dandan Peng

Academic Editor

PLOS ONE

Additional Editor Comments (optional):

Reviewer #2:

Reviewer #3:

Reviewer #4:

Reviewers' comments:

Reviewer's Responses to Questions

**Comments to the Author**

1. If the authors have adequately addressed your comments raised in a previous round of review and you feel that this manuscript is now acceptable for publication, you may indicate that here to bypass the “Comments to the Author” section, enter your conflict of interest statement in the “Confidential to Editor” section, and submit your "Accept" recommendation.

Reviewer #2: All comments have been addressed

Reviewer #3: (No Response)

Reviewer #4: All comments have been addressed

2. Is the manuscript technically sound, and do the data support the conclusions?

Reviewer #2: Yes

Reviewer #3: (No Response)

Reviewer #4: Yes

3. Has the statistical analysis been performed appropriately and rigorously? 

Reviewer #2: N/A

Reviewer #3: (No Response)

Reviewer #4: Yes

4. Have the authors made all data underlying the findings in their manuscript fully available?

Reviewer #2: Yes

Reviewer #3: (No Response)

Reviewer #4: Yes

5. Is the manuscript presented in an intelligible fashion and written in standard English?

Reviewer #2: Yes

Reviewer #3: (No Response)

Reviewer #4: Yes

6. Review Comments to the Author

Reviewer #2: (No Response)

Reviewer #3: The paper can be accepted.

Please use the space provided to explain your answers to the questions above. You may also include additional comments for the author, including concerns about dual publication, research ethics, or publication ethics.

Reviewer #4: Thank you for your thorough revisions in response to the reviewers' comments. After reviewing the manuscript, I believe significant improvements have been made in experimental validation, model comparison, figure quality, and methodological clarity. You have adequately addressed my suggestions, and the quality of the paper has greatly improved.

One small suggestion: Please carefully review the English expressions, as some sentences may not be as natural or clear. Ensuring that the language is polished and fluent before formal publication will further enhance the manuscript’s readability and academic tone.

Therefore, I believe the paper now meets the publication standards and recommend it for acceptance. I look forward to your final version.

7. PLOS authors have the option to publish the peer review history of their article (what does this mean?). If published, this will include your full peer review and any attached files.

Reviewer #2: No

Reviewer #3: No

Reviewer #4: **Yes: **Shilin Wang

---

## [Editor Report · Acceptance letter]

PONE-D-25-34707R1

PLOS ONE

Dear Dr. Chu,

I'm pleased to inform you that your manuscript has been deemed suitable for publication in PLOS ONE. Congratulations! Your manuscript is now being handed over to our production team.

Kind regards,

on behalf of

Dr. Dandan Peng

Academic Editor

PLOS ONE